# Antitumor Effects of Combination Therapy with Oncolytic Vaccinia Virus and Tepotinib on Lung Cancer Cells

**DOI:** 10.3390/cancers17162681

**Published:** 2025-08-18

**Authors:** Takuya Inoue, Nobuhiro Kanaji, Takafumi Nakamura, Masanao Yokohira, Yuta Komori, Yasuhiro Ohara, Hitoshi Mizoguchi, Naoki Watanabe, Norimitsu Kadowaki

**Affiliations:** 1Division of Hematology, Faculty of Medicine, Department of Internal Medicine, Kagawa University, Takamatsu 761-0793, Japan; kanaji.nobuhiro@kagawa-u.ac.jp (N.K.); s23d711@kagawa-u.ac.jp (Y.K.); ohara.yasuhiro@kagawa-u.ac.jp (Y.O.); mizoguchi.hitoshi@kagawa-u.ac.jp (H.M.); watanabe.naoki@kagawa-u.ac.jp (N.W.); kadowaki.norimitsu@kagawa-u.ac.jp (N.K.); 2Division of Genomic Medicine, Department of Genomic Medicine and Regenerative Therapeutics, School of Medicine, Faculty of Medicine, Tottori University, Yonago 683-8503, Japan; taka@tottori-u.ac.jp; 3Department of Medical Education, Faculty of Medicine, Kagawa University, Takamatsu 761-0793, Japan; yokohira.masanao.tm@kagawa-u.ac.jp

**Keywords:** lung cancer, mesenchymal-epithelial transition inhibitor, oncolytic virus, vaccinia virus, virotherapy

## Abstract

Despite recent advances in treatments for advanced lung cancer, survival rates remain low, emphasizing the need for new therapeutic strategies. This study evaluated the combined effects of an oncolytic vaccinia virus, engineered to selectively replicate in cancer cells, and the MET inhibitor tepotinib, known for its ability to enhance immune responses. The combination therapy demonstrated superior antitumor effects compared to each treatment alone, effectively suppressing tumor growth and increasing immune cell infiltration in cell and mouse models of lung cancer. Notably, the therapy not only targeted directly injected tumors but also showed effectiveness against untreated distant tumors. These findings highlight the potential of combining targeted molecular therapies with viral immunotherapy to improve lung cancer treatment outcomes, offering both direct tumor-killing effects and activation of the immune system against cancer.

## 1. Introduction

Molecular-targeted therapies and cancer immunotherapy have significantly improved the prognosis of patients with advanced non-small cell lung cancer (NSCLC). However, the 5-year survival rate in pivotal clinical trials involving immune checkpoint inhibitors and cytotoxic anticancer drugs remains approximately 20% [1,2]. In other words, 80% of patients succumb to the disease within 5 years, highlighting the urgent need for more effective treatment strategies.

Oncolytic virus (OV) therapy utilizes viruses modified to selectively replicate in cancer cells. This selective replication causes tumor cell death and activates a systemic immune response against tumors, highlighting its potential as a therapeutic strategy in oncology [3]. Among OVs, herpes simplex virus-1 (HSV-1) has advanced to clinical use as talimogene laherparepvec (T-VEC), which has been approved for treating malignant melanoma in the United States [4] and for malignant glioma in Japan [5]. Vaccinia virus (VV), belonging to the orthopoxvirus genus within the Chordopoxvirinae subfamily [6], possesses multiple advantageous features that make it a suitable candidate for oncolytic virus therapies. The historical use of VV in smallpox vaccinations underscores its safety profile, supporting its potential use as an oncolytic agent. Moreover, the large genomic capacity of VV enables the incorporation of significant quantities of exogenous DNA without negatively impacting its replication efficiency. Additionally, VV replication is restricted exclusively to the cytoplasm of infected cells throughout its lifecycle [6]. Clinical trials of oncolytic VV have been conducted for liver cancer [7], melanoma [8], prostate cancer [9,10], colorectal cancer [11], and lung cancer [12,13]. In a phase I trial of JX-594, an oncolytic VV derived from the Wyeth strain, stable disease was observed in 3 out of 5 lung cancer patients [12]. Additionally, intrapleural administration of Olvimulogene nanivacirepvec (Olvi-Vec) was found to be safe and elicited a regional immune response in patients with malignant pleural mesothelioma and malignant pleural effusion secondary to metastatic cancer, including lung cancer [13].

Replication of VV relies on the activation of cellular signaling pathways, notably the extracellular signal-regulated kinase (ERK) and major mitogen-activated protein kinase (MAPK) pathways within infected cells. Utilizing this characteristic, we engineered a recombinant vaccinia virus, named MDRVV (MAPK-dependent recombinant vaccinia virus), derived from the Lister strain, exhibiting high selectivity for tumor cells. MDRVV selectively replicates in cancer cells exhibiting constitutive ERK pathway activation [14]. This virus lacks genes encoding the VV growth factor and O1 protein, both of which activate the ERK pathway. Consequently, MDRVV does not proliferate in normal cells, but it does replicate efficiently in cancer cells with permanently activated ERK pathways, offering high tumor-specific proliferative potential. Given the frequent activation of the ERK pathway in lung cancer, particularly in cases with Ras gene mutations, MDRVV is a promising therapeutic candidate for targeting lung cancer.

Mesenchymal-epithelial transition (MET) is the receptor for hepatocyte growth factor (HGF), and its role in promoting lung cancer progression has been well-documented [15]. Lung fibroblasts secrete HGF, which contributes to lung cancer cell proliferation, and MET inhibitors have been shown to suppress tumor growth in mouse models [16]. Amplification of MET is recognized as a key resistance mechanism to epidermal growth factor receptor (EGFR) tyrosine kinase inhibitors [17]. Furthermore, MET exon 14 skipping is a driver mutation in lung cancer, with MET inhibitors showing strong antitumor efficacy [18]. Interestingly, HGF also directly affects immune cells, promoting the migration of immunosuppressive neutrophils to tumor sites and lymph nodes [19] and inducing immunosuppressive dendritic cells [20], thereby dampening antitumor immune responses. MET inhibitors counteract these effects, enhancing antitumor immunity [19]. Therefore, combining MDRVV with tepotinib is expected to provide complementary effects: MDRVV triggers tumor-specific immune activation, while MET inhibition reduces immunosuppressive signals in the tumor microenvironment. Based on these findings, we hypothesize that the combination of a MET inhibitor and MDRVV in lung cancer may exert additive antitumor effects through both direct cytotoxic mechanisms and immune activation. This combination is expected to serve as a potent and rational treatment strategy, offering both direct and indirect antitumor effects. In this study, we investigated the antitumor efficacy of oncolytic VV therapy, MET inhibition, and their combination in lung cancer models.

## 2. Materials and Methods

### 2.1. Reagents

In this study, we utilized the following antibodies: anti-β-actin (#A2228; Sigma-Aldrich, St. Louis, MO, USA), anti-phospho-p44/42 MAPK (Erk1/2) (#9106; Cell Signaling Technology, Beverly, MA, USA), and anti-high mobility group box-1 protein (HMGB-1) (#ab79823; Abcam, Cambridge, UK). The MET inhibitor tepotinib (EMD1214063; #1100598-32-0) was sourced from Cayman Chemical (Ann Arbor, MI, USA) and Selleck Chemicals (Houston, TX, USA).

### 2.2. Cell Culture

Human NSCLC cell lines, including A549 (adenocarcinoma) and EBC1 (squamous cell carcinoma), were purchased from the Japan Cancer Research Bank located in Tokyo, Japan. The mouse lung adenocarcinoma cell line 3LL was sourced from the Institute of Development, Aging, and Cancer at Tohoku University. NSCLC cells were maintained in RPMI-1640 medium containing 10% fetal bovine serum (FBS). The 3LL cell line used here exhibited mild phospho-c-MET expression by Western blot. Since the evaluation of immune responses required immunocompetent mice, we specifically selected the murine lung cancer cell line 3LL rather than human-derived cells that would necessitate immunodeficient models.

### 2.3. Virus Preparation

MDRVV was constructed by deleting viral genes coding for VGF and O1 proteins, followed by the insertion of an expression cassette for luciferase-enhanced green fluorescent protein or Discosoma sp. red fluorescent protein, as previously described [14]. Viral purification and dialysis were performed using OptiPrep™ (Axis-Shield, Oslo, Norway) and Slide-A-Lyzer™ Dialysis Cassettes (Thermo Fisher Scientific, Waltham, MA, USA) per the manufacturer’s protocol. The use of MDRVV was approved by the Ministry of Education, Culture, Sports, Science, and Technology, in accordance with the Act on the Conservation and Sustainable Use of Biological Diversity through Regulations on the Use of Living Modified Organisms (No. 455, 2018).

### 2.4. Cell Viability Assay

The viability of lung cancer cells was evaluated using the WST-8 assay. Cells were briefly incubated in RPMI−1640 medium containing 10% fetal bovine serum (FBS) along with 10% WST-8 reagent (#CK04; Dojindo, Kumamoto, Japan) for 4 h. Subsequently, 100 µL of the mixture from each condition was transferred to individual wells of a 96-well plate, and the absorbance at 450 nm was measured with a microplate reader (iMark^TM^; Bio-Rad, Hercules, CA, USA). All assays were conducted in at least triplicate.

### 2.5. Animal Models

Female C57BL/6J mice were obtained from Charles River Laboratories Japan, Inc. (Yokohama, Japan) and housed at the Division of Animal Experiments, Life Science Research Center, Kagawa University, following the Institutional Regulations for Animal Experiments. All animal experiment protocols were authorized by the Animal Care and Use Committee at Kagawa University (ethical approval no.19601–3). In brief, 2 × 10^6^ 3LL cells were subcutaneously inoculated into the bilateral flanks of mice aged six weeks. Tumorigenesis was observed one week later. MDRVV (1 × 10^7^ PFU in 100 µL PBS) was injected intratumorally into one tumor on days 1 and 6, while EMD1214063 (200 mg in 100 µL PBS) was administered intraperitoneally daily. Control groups received PBS injections instead of the drugs. Additionally, a group in which only one drug was replaced with PBS was included for comparison. Tumor sizes were measured daily with calipers, and tumor volume (TV) was calculated using the formula TV = 1/2 × A × B^2^ (where A is the length in millimeters, and B is the width in millimeters), as described in previous studies [21]. Mice were monitored for up to 33 days post-inoculation, after which they were euthanized. Mice were also euthanized if TV exceeded 2250 mm^3^, approximately 10% of body weight.

### 2.6. Determination of ATP Content

Extracellular adenosine triphosphate (ATP) levels were determined using luminescence assays with the CellTiter-Glo 2.0 Cell Viability Assay (#G9241; Promega, Madison, WI, USA). Briefly, 1 × 10^4^ cells were seeded into each well of a 96-well plate in 200 µL of medium, and MDRVV was added at a multiplicity of infection (MOI) of 1. Supernatants from each well were collected at 24-h intervals, and 100 µL of each supernatant was transferred to a new 96-well plate. Following the addition of 100 µL of CellTiter-Glo reagent, luminescence was measured using an SH-9000 reader (Corona Electric Co., Hitachinaka, Japan). Luminescence values were converted to ATP concentrations using a standard calibration curve.

### 2.7. Histology and Immunohistochemistry

Immunohistochemical staining of tumor sections was performed using monoclonal antibodies specific for CD4 and CD8. Mice were euthanized 14 days after the first viral injection, and tumors were harvested, fixed in formalin, and stained with anti-CD4 antibody (#ab183685; Abcam, Cambridge, UK) and anti-CD8 alpha antibody (#ab209775; Abcam, Cambridge, UK). To quantify positive cells in each tumor, 100 microscopic fields were examined at 400× magnification. All slides were reviewed and scored by a single researcher. For statistical analysis, the number of measurements was reduced to one-tenth of the original count, corresponding to the 10 high-power fields (HPFs) typically used in similar studies.

### 2.8. Western Blotting

After two to three days of viral treatment, cells were collected and lysed in lysis buffer (20 mM Tris-HCl, pH 7.5, 150 mM NaCl, 5 mM EDTA, 0.5% Triton X-100, and 0.5% NP40) containing protease inhibitors (Sigma-Aldrich). Cells were sonicated, and 500 µL of the culture supernatant was collected for analysis. Samples were centrifuged at 14,000 rpm for 5 or 15 min to obtain supernatants. Proteins were separated on 10% SDS-polyacrylamide gels, transferred to nitrocellulose membranes, and blocked with 5% (*w*/*v*) non-fat dried milk in Tris-buffered saline with Tween 20. The membranes were incubated with an anti-HMGB-1 antibody (0.05 µg/mL, #MAB1690, RD 1:2000), followed by horseradish peroxidase-conjugated anti-mouse IgG (Cell Signaling Technology Inc.). Signals were detected using Immobilon Western Chemiluminescent HRP Substrate (Millipore, Billerica, MA, USA).

### 2.9. Real-Time Quantitative PCR

Total DNA was extracted from virus fluids or tumor samples collected from euthanized mice using the QIAamp DNA Mini Kit (Qiagen, Hilden, Germany). Real-time quantitative PCR was performed using TB Green^®^ Premix Ex Taq™ II (#RR820A; Takara Bio Inc., Shiga, Japan) and a ViiA7 (Applied Biosystems, Waltham, MA, USA). Primers were purchased from Takara Bio Inc. PCR amplification involved 40 cycles of denaturation at 95 °C for 5 s, annealing, and elongation at 60 °C for 30 s. Each reaction was performed in duplicate. Viral DNA expression was used as an internal control, and the threshold value (Ct) for each sample was used to determine gene expression levels. A threshold (ΔRn = 0.04) for virus detection was set based on the original virus solution, with a minimum detectable Ct value of 36.149 in the non-injected tumor samples.

### 2.10. Statistical Analysis

Each experiment was repeated at least three times. Student’s *t*-test was generally used for comparing data between two groups. For the analysis of tumor volumes, the Wilcoxon rank-sum test was employed, while survival analysis was conducted using the Generalized Wilcoxon test. Data are expressed as means ± standard error (SE). *p*-values less than 0.05 were considered statistically significant. All statistical analyses were conducted using JMP 17.0.0 (SAS Institute, Cary, NC, USA).

## 3. Results

### 3.1. Cytotoxic Effects of MDRVV and Tepotinib on Lung Cancer Cell Lines

We assessed the cytotoxic effects of MDRVV and tepotinib on the lung cancer cell lines A549, EBC-1, and 3LL. Cell numbers were counted after 3 days of culture with varying concentrations of both agents. Despite differences in sensitivity, all cells underwent cell death upon treatment with either MDRVV or tepotinib. The combination of both agents produced an additive cytotoxic effect (Figure 1). Phosphorylation of ERK1/2 was detected by Western blotting in all three cell lines.

### 3.2. ATP Concentration in Culture Supernatant and HMGB-1 Expression

To investigate whether MDRVV induced immunogenic cell death, we measured ATP concentrations in culture supernatants and assessed HMGB-1 expression in A549 cells. Cisplatin was used at a concentration that resulted in approximately 50% mortality after 3 days of exposure (20 µM). MDRVV was added at an MOI of 10. After MDRVV administration, ATP concentration in the culture supernatant peaked at 72 to 96 h (Figure 2A). No increase in ATP concentration was observed after treatment with either tepotinib or cisplatin (Figure 2B). The supernatant was collected 3 days after MDRVV treatment of A549 cells, and HMGB-1 release was confirmed using Western blotting (Figure 2C). Elevated ATP and HMGB-1 levels suggest that MDRVV induces immunogenic cell death.

### 3.3. Combination Treatment Inhibits Tumor Progression in Mouse

To investigate whether MDRVV exerts an immune-mediated antitumor effect, we used C57BL/6 mice with an intact immune system. We inoculated 2 million 3LL cells subcutaneously into two sites on the backs of the mice. Tumorigenesis was observed one week later. MDRVV was injected intratumorally on days 1 and 6 in only one tumor. Tepotinib was administered daily via intraperitoneal injection. The combination treatment significantly inhibited tumor growth in both the MDRVV-injected and non-injected tumors compared to the control group (Figure 3A,B). Kaplan–Meier survival curves based on euthanasia due to tumor growth revealed significant survival advantages for both MDRVV and tepotinib monotherapy compared to the control group (both *p* < 0.05, Figure 3C). There was no significant difference between the MDRVV and tepotinib groups (*p* = 0.88). The combination treatment group exhibited significantly longer survival than both the MDRVV or tepotinib monotherapy groups (both *p* < 0.05) and the control group (*p* < 0.001, Figure 3C).

### 3.4. Detection of Vaccinia Virus in MDRVV-Injected and Non-Injected Tumors

Real-time PCR revealed that vaccinia virus was undetectable in the control group. In the virus-injected tumors, vaccinia virus was detected in seven out of eight mice, while in the non-injected tumors, it was detected in five out of eight mice.

### 3.5. Pathological Analysis of Tumors After Treatment

Hematoxylin-eosin staining of mouse tumor samples did not reveal any obvious differences in lymphocyte infiltration between the groups. To investigate the immunomodulatory effects of MDRVV and tepotinib, we performed immunohistochemical staining for CD4 and CD8 (Figure 4). In both injected and non-injected tumors, the combination group showed higher CD4+ and CD8+ cell infiltration compared to the single-agent groups. Specifically, CD8+ cell infiltration was significantly higher in the combination group, suggesting a stronger immune response.

## 4. Discussion

In this study, we demonstrated that (1) MDRVV and tepotinib exhibit antitumor effects on lung cancer cell lines both in vitro and in vivo, with additive effects when combined. In our study, the IC50 of tepotinib in EBC-1 cells was measured as 388 nM, which is higher than the nanomolar-range values (1–9 nM) reported in previous studies [22]. Such differences may reflect variations in assay conditions, passage number, or measurement endpoints, and should be interpreted with caution when comparing across studies. (2) MDRVV induces immunogenic cell death, as evidenced by the release of ATP and HMGB-1; and (3) MDRVV promotes antitumor effects and induces infiltration of CD4+ and CD8+ T-cells not only in inoculated tumors but also in distant tumors, with tepotinib enhancing these effects. Immune checkpoint inhibitors (ICIs), such as anti-programmed cell death-1 (PD-1) antibodies, have proven effective against “hot tumors,” which are characterized by pre-existing immune cell infiltration. However, their efficacy is reduced in “cold tumors,” where immune responses are limited [23]. The goal of “combination cancer immunotherapy” is to convert cold tumors into hot ones. One promising approach to achieve this is through the use of oncolytic viruses [24,25]. OVs not only directly target and kill tumor cells but also disrupt the immunosuppressive tumor microenvironment, allowing immune cells to mount a robust antitumor response. OVs activate dendritic cells, which phagocytose dead tumor cells and present tumor antigens, including neoantigens, to trigger antitumor T-cell responses [26]. This sequence of immune responses is referred to as the “cancer-immunity cycle” [27]. Although ATP and HMGB-1 release are established markers of immunogenic cell death, further confirmation through additional assays such as calreticulin exposure or dendritic cell activation could strengthen these conclusions in future studies. When immunogenic cell death is induced in tumor cells, damage-associated molecular patterns (DAMPs) and various tumor antigens are released [28,29]. In our study, MDRVV induced the release of ATP and HMGB-1, both representative DAMPs, in lung cancer cell lines. Thus, MDRVV was shown to induce immunogenic cell death and increase CD4+ and CD8+ T-cell infiltration in the tumor microenvironment, suggesting that it promotes the cancer-immunity cycle. Additionally, tepotinib may enhance therapeutic efficacy by recruiting lymphocytes, in addition to its direct antitumor effects [19]. Importantly, our results showed that significantly more CD8+ T-cells accumulated in tumor tissues when MDRVV and tepotinib were used in combination compared to either agent alone. This supports the hypothesis that lymphocytes activated by MDRVV more readily migrate into the tumor microenvironment when tepotinib is administered. Thus, the combination of these agents enhances antitumor efficacy through complementary mechanisms that amplify immune responses. In this study, we interpreted the combined effects as “additive” rather than strictly synergistic. A formal synergy analysis, such as the Chou-Talalay method, could be valuable in future investigations.

In this study, MDRVV was injected directly into subcutaneous tumors. However, lung cancer metastases are not always located in the skin or superficial lymph nodes. Interestingly, MDRVV was detected in 62.5% of non-injected tumors. VV is known for its ability to efficiently reach tumors through the bloodstream while evading host immunity—a property distinct from that of HSV-1, which is rapidly cleared when administered intravenously [30]. In a phase I study, VV (Pexa-Vec) was shown to dose-dependently affect various types of cancer following a single intravenous administration, without affecting normal tissues [12]. Therefore, if direct tumor injection is not feasible, intravenous administration of VV may still be an effective treatment for advanced lung cancer.

Several studies have explored VV variants with enhanced antitumor effects. Fusogenic oncolytic VV was shown to induce immunogenic cell death and exhibited superior antitumor activity compared to non-fusogenic virus [31]. VV encoding interleukin-7 (IL-7) and IL-12 have been shown to activate immune responses, leading to complete tumor regression [32]. Additionally, VV that drives IL-9 gene expression demonstrated prolonged viral persistence in tumors and increased CD4+ and CD8+ T-cell infiltration in the tumor microenvironment in mice [33]. ICIs are currently the standard of care for advanced lung cancer. Given its immunostimulatory properties, OV therapy is an excellent candidate for combination with ICIs, as demonstrated in previous studies [31,32]. In the near future, the combination of ICIs and oncolytic viruses, particularly VV, may become a cornerstone in the treatment of lung cancer. Although no overt toxicities were observed, potential combination toxicities between the virus and MET inhibitor should be carefully evaluated in future preclinical safety studies.

This study has some limitations. First, the immune responses induced by MDRVV and tepotinib were not thoroughly examined, except for CD4+ and CD8+ T-cell infiltration. Although PD-L1 expression is currently a predictor of efficacy for anti-PD-1/PD-L1 antibodies, the most commonly used ICIs for lung cancer, we did not investigate changes in PD-1 expression following MDRVV administration. Second, the administration method and frequency of MDRVV were not fully explored. We did not attempt intravenous administration, and it remains unknown whether increasing the frequency of MDRVV administration could yield comparable antitumor efficacy to that achieved with combination therapy with tepotinib. Third, detailed immune profiling, including flow cytometry and cytokine assays, was not performed in this study; future studies should incorporate these analyses as an important direction for elucidating the underlying immune mechanisms.

In addition, while MDRVV was detected in distant, non-injected tumors, supporting its systemic distribution and antitumor activity, the pharmacokinetics and delivery efficiency to metastatic or poorly perfused lesions remain to be fully elucidated and warrant further investigation. The potential for off-target immune activation or combined toxicities with viral and targeted therapies was not fully evaluated. Comprehensive safety assessments will be essential before clinical translation. Finally, although the combination therapy demonstrated clear antitumor activity and delayed tumor progression, durable tumor control was not achieved in this study. Evaluating the durability of the immune response induced by this therapy is an important next step, as long-term immune memory is crucial for sustained clinical benefit. Future studies should include long-term monitoring to evaluate immune memory and strategies to achieve durable tumor remission.

## 5. Conclusions

This study demonstrated that oncolytic vaccinia virus and tepotinib inhibit lung cancer cell proliferation in vitro and suppress tumor progression in vivo. The combination therapy exhibited an additive antitumor effect, combining direct cytotoxic actions with indirect immunostimulatory effects. These findings suggest that this combination therapy holds promise for clinical applications, and future clinical applications are being planned.

## Figures and Tables

**Figure 1 cancers-17-02681-f001:**
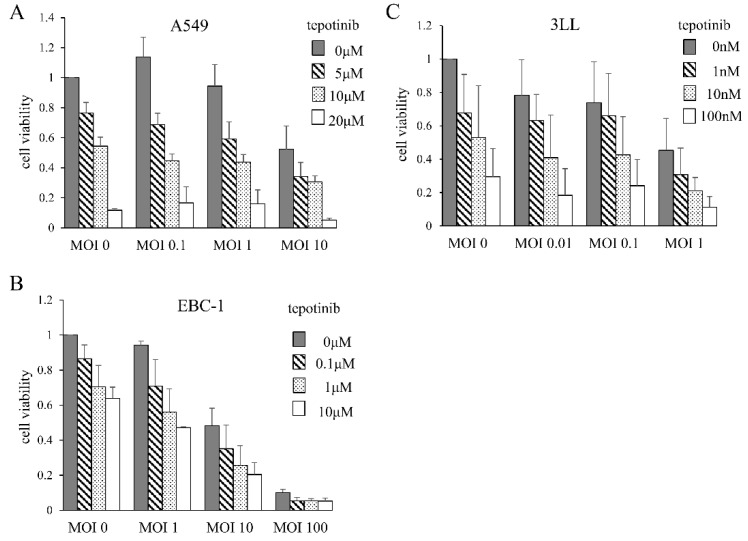
Cytotoxic effects of MDRVV and tepotinib on lung cancer cell lines. Cells were cultured for 3 days with varying concentrations of MDRVV and/or tepotinib, and then cell numbers were counted. Cell viability was calculated with neither MDRVV nor tepotinib as 1.0. (**A**) A549: human lung adenocarcinoma cell line. (**B**) EBC-1: human lung squamous cell line. (**C**) 3LL: mouse lung adenocarcinoma cell line. MOI: multiplicity of infection.

**Figure 2 cancers-17-02681-f002:**
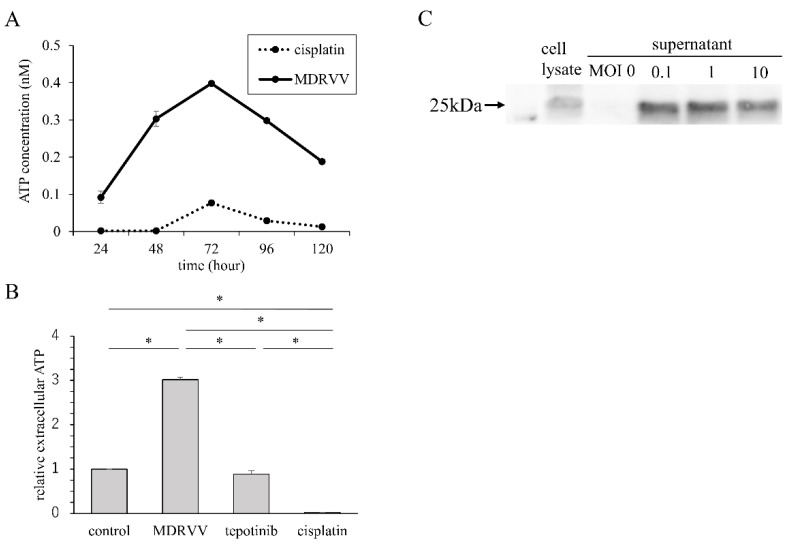
Increase in molecules associated with immunogenic cell death in A549 cells. (**A**) Time course of ATP concentration. Cells were cultured with MDRVV or cisplatin, and ATP concentrations in the supernatant were measured daily. (**B**) ATP concentrations in the supernatant compared to control after 72 h of exposure with MDRVV, tepotinib, or cisplatin. * *p* < 0.05 compared to each agent. (**C**) HMGB-1 expression. Cells were cultured for MDRVV for 3 days, and cell lysate and supernatant were collected. HMGB-1 expression was assessed using Western blotting.

**Figure 3 cancers-17-02681-f003:**
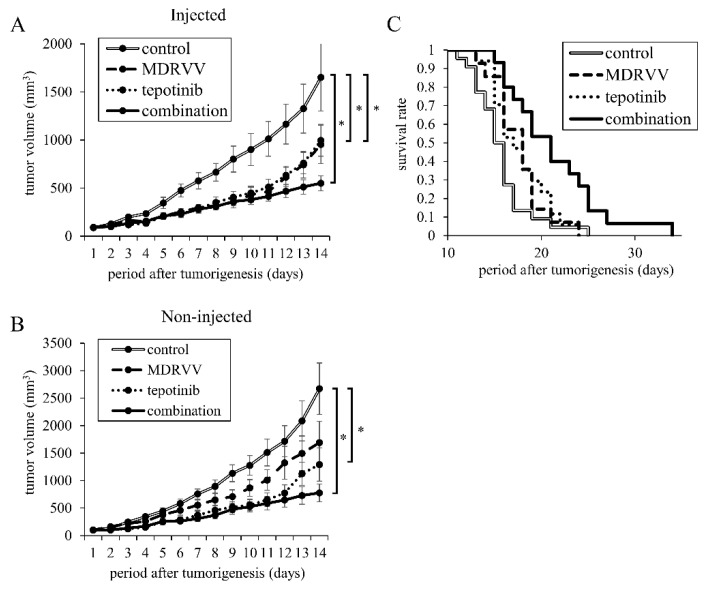
Effects of MDRVV and tepotinib on tumor progression and survival in mice. 3LL cells were subcutaneously inoculated in two sites per mouse. After tumorigenesis, MDRVV were injected into one tumor. Tepotinib was administered intraperitoneally. (**A**,**B**) The volumes of (**A**) MDRVV-injected tumor and (**B**) MDRVV-non-injected tumor. The number of mice used was 10 for the combination group and 9 for the other groups. * *p* < 0.05 for comparison of both groups. (**C**) Kaplan–Meier curves for survival in four groups. When the tumor volume exceeded 2250 cubic millimeters, which is approximately 10% of the body weight, mice were euthanized. The number of mice used was 22, 14, 17, and 15 in the control, MDRVV, tepotinib, and combination groups, respectively. The combination group survived significantly longer than all groups (*p* < 0.05).

**Figure 4 cancers-17-02681-f004:**
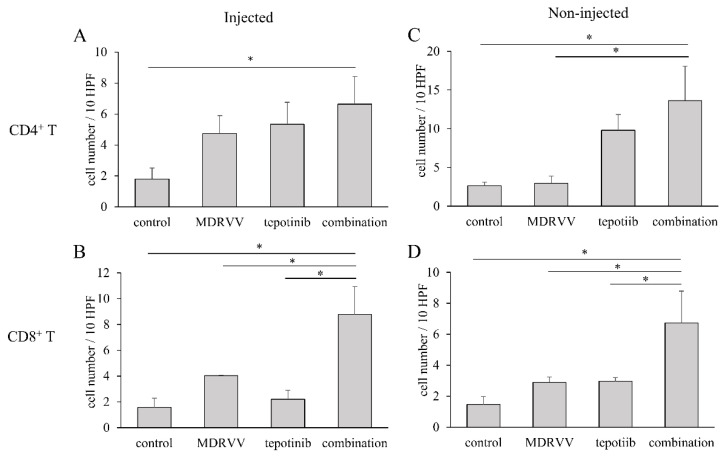
Tumor-infiltrating lymphocytes. Tumors were resected from euthanized mice, as shown in Figure 3C, and immunostainings for CD4 and CD8 were performed. (**A**,**B**) MDRVV-injected tumors. (**C**,**D**) MDRVV-non-injected tumors. (**A**,**C**) CD4-positive T-cells. (**B**,**D**) CD8-positive T-cells. * *p* < 0.05 for comparison of both groups.

## Data Availability

The data presented in this study are available from the corresponding author on reasonable request due to the large number of instrument-specific raw files and the time required for curation.

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
