# Peer review of "Antitumor Effects of Combination Therapy with Oncolytic Vaccinia Virus and Tepotinib on Lung Cancer Cells"

_cancers, 2025, doi:10.3390/cancers17162681_

Round 1

Reviewer 1 Report

Comments and Suggestions for Authors

This manuscript describes experiments combining cMET tyrosine kinsase inhibitor in combination with oncolytic vaccinia virus in preclinical models of NSCLC.  Here the main idea is that cMET plays a role in disease progression and also inhibition of cMET can recruit immune effectors into the tumor microenvironment potentially leading to enhanced antitumor activity. 

The idea is somewhat interesting and to my knowledge has not been tested before. Overall the effects of combination therapy is rather modest in all of the cell lines tested.  The authors use a Met-amplified NSCLC cell line EBC-1, A549 which is a Kras Mutated cell line that expresses Met, and 3LL which also expressed Met.  

Surprisingly none of these cell lines are particularly sensitive to tepotinib, especially the EBC-1 which has been shown to be sensitive to tepotinib in prior published work in the nanomolar range.  Here we see IC50 of tepotinib not reached even at 10uM.  Both A549 and 3LL cells are more sensitive to tepotinib which is not addressed here.  

Likewise, these cells are not very sensitive to MDRVV either in vitro or in vivo. The authors state that these are additive effects but I am not sure we see that much improvement in efficacy in vitro until we get to high doses of both.  A formal analysis of the combination is called for here - like Combination Index to better assess. 

It would also be useful to test a cMET sensitive cell line - one that has ex14 skipping mutation,e.g., to see what would happen in a very sensitive cell line, such as H596 cell line. 

The animal experiments show virtually no antitumor activity of MDRVV by itself, and only modest improvements after combination therapy both in tumor volume and in survival.  There is no significant improvement in combination therapy compared to tepotinib alone.  Thus the value of this combination is a bit suspect.  

The CD8/CD4 assessment of tumors is interesting noting increase in CD8 and CD4 t cell infiltration with the combination - however, It would be useful to describe the methodology better - For example, when were the tumors analyzed for T cell infiltration? How representative are the slides to the whole tumor - FLow cytometry of dissociated tumors would be a better way to describe this. 

What antibodies were used for the IHC - maybe better to validate the findings by analysis by a blinded pathologist, eg. 

Reviewer 2 Report

Comments and Suggestions for Authors

This manuscript is a well-written and timely study demonstrating the anti-tumor effects of combination of MET inhibitor (tepotinib) and an oncolytic vaccinia virus (MDRVV) in lung cancer. The study provides important insights into the immunomodulatory synergy of this combination therapy.

However, the following points should be addressed.

While the authors showed potential synergistic effects between MDRVV and tepotinib through cytotoxic and immunomodulatory mechanisms, the rationale for selecting this specific combination should be described in more detail. I suggest that the authors strengthen the Introduction section by more clearly describing why MET inhibition complements the oncolytic vaccinia virus.

The authors evaluated ATP and HMGB-1 release to characterize immunogenic cell death induced by MDRVV. Is this sufficient to show immunogenic cell death? Please discuss this issue.

The authors utilized 3LL murine lung cancer cells to evaluate immune responses. This is an appropriate model for studying tumor-immune interactions. Could the authors use alternative murine lung cancer models? Does 3LL cells highly express MET? I think it is interesting to clarify the effect of this combination therapy in MET-low or MET-negative cell lines.

While the authors demonstrate enhanced CD4+/CD8+ T cell infiltration, further mechanistic analysis of the immune landscape such as flow cytometry or cytokine profiling would strengthen the findings.

Reviewer 3 Report

Comments and Suggestions for Authors

This research investigates the effects of combining a modified vaccinia virus (MDRVV) used in oncolytic virotherapy with tepotinib, a MET pathway inhibitor, to advance lung cancer treatments. Traditional targeted therapies have yet to significantly improve survival in advanced cases, emphasizing the need for innovative methods. The dual treatment showed enhanced cytotoxicity in vitro and substantially curtailed tumor growth in immune-competent mouse models. MDRVV triggered immunogenic cell death, releasing molecules such as ATP and HMGB-1, while the combination regimen promoted greater infiltration of CD4+ and CD8+ T cells compared to single agents. Additionally, MDRVV displayed antitumor activity at both local and distant sites, effects that were strengthened when paired with tepotinib, offering an effective approach to target tumors and their microenvironment simultaneously. following are my observations for improvement of the work;

  1. Kindly check that all the abbreviations are defined during the first instance of usage.
  2. Ensure that punctuation and grammatical errors are corrected throughout the manuscript.
  3. What about bioavailability and delivery challenges for this?
  4. What about off-target immune activation and other side effects?
  5. Discuss the study limitations properly with future prospects.
  6. What about combination toxicities i.e, virus and drug
  7. Durability of immune response
  8. Figure 2C is not properly visible
  9. Authors are requested to check the referencing as per journal requirements

Round 2

Reviewer 2 Report

Comments and Suggestions for Authors

I sincerely appreciate your thoughtful and sincere responses to the reviewers' comments. I have a strong interest in the immunological analysis and understand that it is planned for future studies.